# VASCULAR INFORMATION-GUIDED AUTOMATED CONVERSION BETWEEN OCT AND OCTA

## ABSTRACT

Optical Coherence Tomography (OCT) and Optical Coherence Tomography Angiography (OCTA) provide complementary perspectives for retinal disease assessment—OCT captures structural layers while OCTA visualizes microvascular networks. However, their synergistic use is limited by OCTA's specialized equipment requirements and higher costs, particularly in resource-constrained settings. Moreover, clinical scenarios often require cross-validation between modalities to verify vascular-structural correlations or to compensate for artifacts and missing data.

We propose a bidirectional conversion framework that enables seamless transformation between OCT and OCTA modalities, facilitating comprehensive multimodal analysis from single-modality inputs. Our approach integrates generative adversarial networks with wavelet decomposition and attention mechanisms to preserve both morphological coherence and diagnostic fidelity. The framework consists of three synergistic components: a 3D Cross-Modal Transformer for volumetric synthesis, a Vessel Structure Matcher for vascular topology preservation, and Hierarchical Feature Calibration for layer-specific refinement.

Extensive validation on the OCTA-500 dataset demonstrates superior performance with PSNR of 30.58, SSIM of 90.64%, and MAE of 0.0199 for OCT-to-OCTA synthesis. Clinical disease classification experiments show that combining real OCT with synthesized OCTA achieves 75.06% accuracy, surpassing single-modality baselines by up to 29%. This bidirectional capability not only addresses accessibility barriers but also enables cross-modal validation essential for artifact disambiguation and comprehensive diagnostic assessment, ultimately advancing precision medicine in ophthalmology.The code is available in the supplementary materials.

## 1 INTRODUCTION

Optical Coherence Tomography (OCT) has emerged as an essential imaging modality in clinical ophthalmology, enabling non-invasive acquisition of high-resolution three-dimensional retinal structures. This technology has become indispensable for diagnosing and monitoring major ophthalmic pathologies, including age-related macular degeneration, diabetic retinopathy, and glaucoma (Huang et al., 1991; Fujimoto & Swanson, 2016; Hammes et al., 2011; Hong et al., 2024). Building upon conventional OCT, Optical Coherence Tomography Angiography (OCTA) provides non-invasive visualization of retinal microvasculature and hemodynamic characteristics, offering valuable clinical information for detecting pathological changes such as choroidal neovascularization (Kashani et al., 2017; Jiang et al., 2020; Ting et al., 2017). Despite their complementary diagnostic capabilities, widespread adoption of high-quality OCTA imaging remains constrained by specialized hardware requirements, susceptibility to noise and motion artifacts, and substantially higher equipment costs.

The ability to perform bidirectional conversion between OCT and OCTA modalities carries profound implications for multi-modal medical image analysis. Beyond addressing accessibility constraints, bidirectional synthesis enables comprehensive diagnostic workflows where each modality's strengths compensate for the other's limitations. OCT-to-OCTA synthesis democratizes access to vascular imaging in resource-limited settings, while OCTA-to-OCT conversion provides crucial structural context for vascular findings, facilitates artifact disambiguation, and enables validation

when original structural images are compromised or unavailable. This bidirectional capability transforms single-modality acquisitions into multi-modal diagnostic resources, enabling clinicians to correlate vascular abnormalities with structural alterations, verify suspicious findings across modalities, and maintain diagnostic continuity even when one imaging modality is affected by artifacts or technical limitations.

Recent advances in deep learning have demonstrated promise for cross-modal medical image synthesis (Jiang et al., 2020). However, existing research has predominantly focused on unidirectional OCT-to-OCTA generation, overlooking the inverse transformation's clinical value. The OCTA-to-OCT pathway offers unique diagnostic advantages: it helps distinguish true vascular signals from projection artifacts, provides anatomical landmarks for localizing vascular pathology, and enables comprehensive assessment when structural imaging is suboptimal. Furthermore, bidirectional synthesis facilitates iterative refinement and cross-validation between modalities, enhancing diagnostic confidence through mutual information reinforcement.

In this work, we present a comprehensive framework that achieves reliable bidirectional conversion between OCT and OCTA modalities with the following key contributions:

- **Novel bidirectional synthesis framework:** We propose the first comprehensive architecture for both OCT-to-OCTA and OCTA-to-OCT transformation, enabling true multi-modal synergistic analysis from single-modality inputs. This bidirectional capability leverages the complementary strengths of structural and functional imaging to maximize diagnostic information extraction.

- **Advanced morphological preservation mechanisms:** We integrate wavelet-based frequency decomposition with multi-scale attention mechanisms and introduce specialized modules—Vessel Structure Matcher and Hierarchical Feature Calibration—to ensure morphological coherence, vascular topology preservation, and layer-specific structural fidelity essential for clinical reliability.

- **Superior synthesis performance with clinical validation:** Our framework achieves state-of-the-art performance (PSNR: 30.58, SSIM: 90.64%, MAE: 0.0199) on the OCTA-500 dataset. Clinical validation through disease classification demonstrates that synthesized modalities enhance diagnostic accuracy by up to 29% compared to single-modality baselines, confirming practical clinical utility.

Figure 1 illustrates our complete pipeline. By enabling comprehensive multi-modal analysis from single acquisitions, this research advances precision medicine in ophthalmology, potentially improving patient care while reducing healthcare costs and expanding diagnostic capabilities in resource-limited settings.

## 2 RELATED WORK

### 2.1 OCT AND OCTA

Optical Coherence Tomography (OCT) and Optical Coherence Tomography Angiography (OCTA) are foundational imaging modalities in modern ophthalmology, offering complementary views of retinal health. OCT utilizes low-coherence interferometry to generate high-resolution, depth-resolved cross-sectional images, providing unparalleled visualization of retinal anatomical structures and pathological changes such as edema or atrophy. However, its capacity to delineate fine vascular structures is inherently limited due to low contrast between blood vessels and surrounding tissue (Bouma et al., 2022).

Conversely, OCTA excels at visualizing vascular networks by detecting the motion of red blood cells, enabling detailed, non-invasive mapping of blood flow in retinal and choroidal plexuses down to the capillary level (Le et al., 2024). This functional information is critical for diagnosing and managing vascular diseases. Despite its strengths, OCTA is susceptible to imaging artifacts and background noise, and its high equipment cost and operational complexity hinder its universal adoption (Chua et al., 2024). The complementary relationship between OCT's structural detail and OCTA's vascular information creates a compelling clinical motivation for developing methods that can synthesize one modality from the other, thereby maximizing diagnostic information from available data.

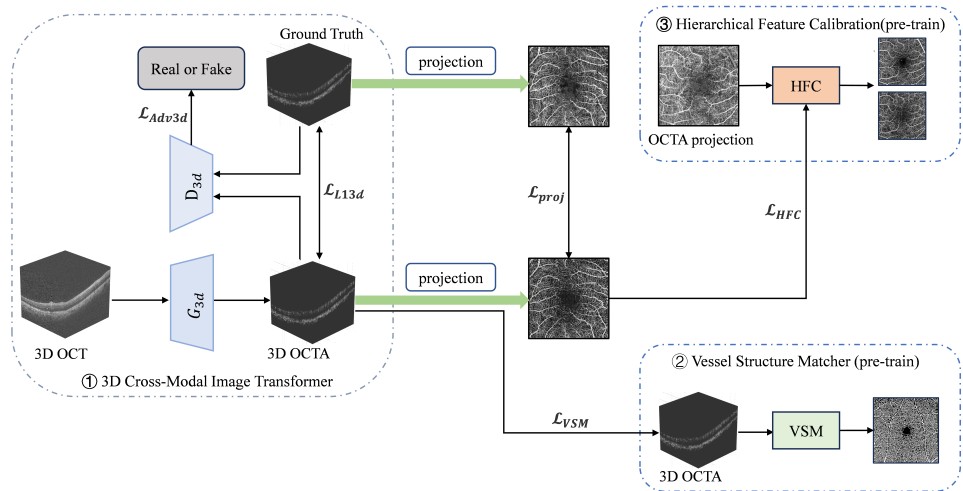

Figure 1: The overall architecture and key components of the proposed framework. **(1)** The 3D Cross-Modal Transformer uses wavelet-attention for synthesis and preserves anatomical and vascular structures. **(2)** The Vessel Structure Matcher (VSM) ensures vascular topology consistency via segmentation validation. **(3)** The Hierarchical Feature Calibration (HFC) performs layer-specific refinement across retinal depths.

## 2.2 CROSS-MODAL IMAGE CONVERSION TECHNIQUES IN MEDICAL IMAGING

Cross-modality medical image synthesis aims to generate images of one modality from another, offering potential to reduce radiation exposure, lower costs, and provide complementary information without additional scans.

Generative Adversarial Networks (GANs) (Goodfellow et al., 2014) have emerged as a significant methodology for synthesizing medical images. Early approaches in this domain primarily focused on 2D image synthesis. For instance, AdjacentGAN (Li et al., 2020) proposed using a stack of three adjacent OCT B-scans to generate the central OCTA B-scan, leveraging local spatial context to improve image continuity. Similarly, MultiGAN (Pan et al., 2022) operated on 2D projection maps, generating en-face OCTA images of different retinal layers from corresponding OCT projections. While these 2D methods demonstrated feasibility, they inherently struggle to model the complex, continuous 3D vascular architecture that traverses multiple retinal layers, a limitation that can compromise the global anatomical accuracy of the generated vasculature.

Recognizing this limitation, subsequent research shifted towards 3D volumetric synthesis. TransPro (Li et al., 2024b) pioneered processing entire 3D OCT volumes to generate corresponding 3D OCTA volumes, improving spatial coherence and structural integrity. Recently, MuTri (Chen et al., 2025) proposed a discrete-space approach using VQ-VAE with multi-view alignment, yet it remains confined to unidirectional OCT-to-OCTA synthesis.

However, significant challenges persist in current approaches. Existing literature predominantly focuses on unidirectional OCT-to-OCTA synthesis, lacking the morphological fidelity mechanisms essential for reliable clinical diagnostics. Moreover, the reverse transformation from OCTA to OCT remains largely unexplored, limiting the potential for comprehensive multimodal analysis.

To address these critical limitations, we propose a novel bidirectional synthesis framework that leverages the complementary strengths of both OCT and OCTA modalities. Our method innovatively integrates wavelet transformation into the generative process, decomposing data into distinct frequency bands to simultaneously capture large-scale anatomical structures (low frequencies) and fine-grained vascular details (high frequencies). This bidirectional approach enables mutual enhancement between modalities—OCT provides structural context to guide vascular reconstruction, while OCTA offers perfusion information to enrich anatomical synthesis.

Furthermore, we employ multi-scale attention mechanisms to preserve critical local features and introduce adaptive augmentation strategies for enhanced robustness. This synergistic bidirectional framework not only produces visually coherent synthesized images but also ensures structural and

morphological reliability crucial for clinical evaluation. By allowing seamless translation between both modalities, our approach facilitates comprehensive disease diagnosis through complementary information fusion, ultimately improving diagnostic accuracy and clinical decision-making.

## 3 APPROACH

Our proposed methodology comprises three synergistic components designed to achieve clinically reliable bidirectional conversion between OCT and OCTA modalities. Each component addresses specific challenges in cross-modal synthesis: the 3D Cross-Modal Transformer handles volumetric transformation while preserving spatial coherence, the Vessel Structure Matcher ensures vascular morphological fidelity, and the Hierarchical Feature Calibration maintains layer-specific structural accuracy. Together, these components form a comprehensive framework for multi-modal retinal image analysis.

### 3.1 3D CROSS-MODAL IMAGE TRANSFORMER

We introduce WaveAtten3D-GAN, a generative framework that achieves high-fidelity bidirectional synthesis between OCT and OCTA volumetric modalities. Building upon the pix2pix architecture (Isola et al., 2017), our method employs a fully three-dimensional convolutional network that processes entire retinal volumes holistically, thereby preserving inter-slice continuity and anatomical coherence—effectively addressing the spatial discontinuities that plague traditional 2D slice-wise approaches.

The architectural innovation lies in the seamless integration of wavelet-based frequency decomposition with attention-guided reconstruction. This dual mechanism enables hierarchical feature learning: the wavelet transformation disentangles retinal volumes into distinct frequency domains, where low-frequency components encode fundamental tissue layer organization while high-frequency components capture intricate microvascular patterns and subtle pathological features. Complementing this decomposition, our multi-scale attention module selectively emphasizes clinically relevant structures during bidirectional synthesis, ensuring that both macroscopic anatomical landmarks and microscopic vascular networks are faithfully preserved.

This synergistic design enables our framework to leverage the complementary diagnostic information from both modalities—structural depth from OCT and perfusion dynamics from OCTA—thereby enhancing the synthesis quality in both transformation directions. The bidirectional capability not only enriches the available imaging data for comprehensive disease assessment but also facilitates cross-validation between modalities, ultimately improving diagnostic confidence. The complete generator architecture is illustrated in Figure 2.

To capture both global anatomical structures and fine-grained vascular details, we integrate two complementary mechanisms:

**Hierarchical Feature Decomposition via 3D Wavelet Transform** (Phung et al., 2023): We employ a 3D Discrete Wavelet Transform (DWT) module to achieve hierarchical multi-resolution analysis of volumetric features. The 3D-DWT decomposes input feature maps $F_{in} \in \mathbb{R}^{C \times D \times H \times W}$ through frequency-domain separation, simultaneously capturing global anatomical structures in low-frequency approximations and preserving fine-grained details across multiple spatial orientations in high-frequency components. This explicit decomposition is particularly crucial for maintaining delicate microvascular networks that predominantly manifest in high-frequency domains. After specialized processing, the inverse wavelet transform reconstructs an enhanced feature representation that maintains both structural integrity and vascular detail (Figure 3).

**Dynamic Fusion and Multi-Scale Attention** (Ouyang et al., 2023): To adaptively integrate multi-resolution information and capture long-range dependencies, we introduce a two-stage refinement mechanism. First, wavelet-reconstructed features are dynamically fused with encoder features through a learnable gating mechanism:

$$F_{fused} = \sigma(\beta) \cdot F_{enc} + (1 - \sigma(\beta)) \cdot F_{wave} \tag{1}$$

where $\beta$ is a learnable gating parameter, $\sigma(\cdot)$ denotes the sigmoid activation function that controls the balance between original feature space $F_{enc}$ and wavelet-enhanced space $F_{wave}$. Subsequently,

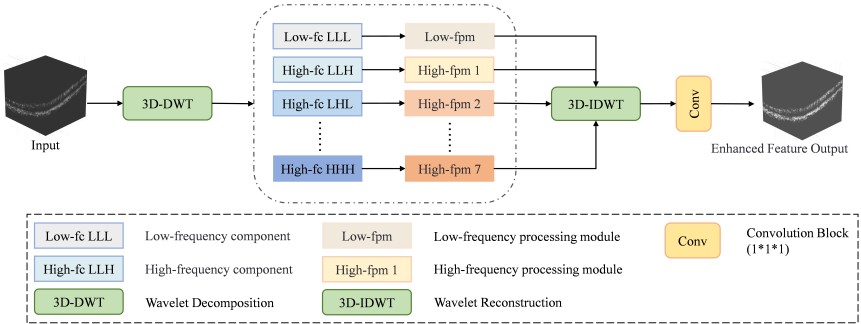

Figure 2: Architecture of the WaveAtten3D generator.The encoder-decoder uses progressive downsampling (N/2→N/256) and channel expansion (d→8d), with wavelet modules for hierarchical frequency decomposition; skip connections preserve high-resolution features for volumetric OCT-OCTA synthesis.

Figure 3: Wavelet module architecture for multi-resolution feature decomposition and reconstruction.

an Efficient Multi-scale Attention (EMA) module refines the fused features through adaptive recalibration:

$$F_{out} = (1 - \alpha) \cdot F_{fused} + \alpha \cdot \mathcal{A}_{EMA}(F_{fused}) \tag{2}$$

where $\mathcal{A}_{EMA}$ represents the EMA operator that models cross-scale spatial dependencies, and $\alpha$ is a learnable intensity parameter controlling the degree of attention-based refinement. This mechanism ensures comprehensive capture of both local vascular details and global anatomical context, essential for accurate cross-modal synthesis.

To enhance training stability and synthesis quality, we incorporate Adaptive Discriminator Augmentation (ADA) (Zhao et al., 2020), which dynamically adjusts augmentation intensity based on training dynamics, preventing discriminator overfitting and enabling more nuanced feature learning.

Our framework processes complete 3D volumes through generator $G_{3d}$ and discriminator $D_{3d}$, where $X \sim p(\text{OCT})$ and $Y \sim p(\text{OCTA})$ represent volumetric data from respective modalities. The adversarial loss ensures realistic synthesis:

$$\mathcal{L}_{Adv3d} = \mathbb{E}_{\mathbf{Y} \sim p(\text{OCTA})} \left[ \log \left( D_{3d} \left( \mathbf{Y} \right) \right) \right] + \mathbb{E}_{\mathbf{X} \sim p(\text{OCT})} \left[ \log \left( 1 - D_{3d} \left( G_{3d} \left( \mathbf{X} \right) \right) \right) \right] \tag{3}$$

Additionally, L1 loss constrains pixel-level consistency to preserve structural fidelity:

$$\mathcal{L}_{L13d} = \| Y - G_{3d}(X) \|_1 \tag{4}$$

The total loss for the WaveAtten3D-GAN is:

$$\mathcal{L}_{3dGAN} = \mathcal{L}_{Adv3d} + \lambda_A \mathcal{L}_{L13d} \tag{5}$$

## 3.2 VESSEL STRUCTURE MATCHER (VSM)

Accurate synthesis of OCTA images with intricate vascular details requires stringent structural consistency between generated and target volumes. To address this critical requirement, we introduce the Vessel Structure Matcher (VSM), a specialized module designed to ensure morphological coherence and topological integrity of vascular networks. VSM validates that essential vascular characteristics—including connectivity patterns, branching topology, and overall architectural organization—faithfully match the reference anatomy. This verification mechanism substantially enhances the authenticity and precision of synthesized vasculature, thereby improving clinical interpretability and diagnostic reliability.

We pre-trained a vessel segmentation model based on the IPN-V2 architecture (Li et al., 2024a) to extract 2D pixel-level vessel segmentation maps from 3D OCTA volumes. Following successful training, we employ L1 loss to quantify structural fidelity by minimizing pixel-wise discrepancies between real and synthesized vessel networks. Let $x_{OCTA}$ and $\widehat{x}_{OCTA}$ denote real and generated 3D OCTA volumes, respectively:

$$\mathcal{L}_{VSM} = \mathbb{E}_{x_{OCTA}, \widehat{x}_{OCTA}} \left\| G_{VSM}(x_{OCTA}) - G_{VSM}(\widehat{x}_{OCTA}) \right\|_1 \tag{6}$$

## 3.3 HIERARCHICAL FEATURE CALIBRATION (HFC)

Complementing VSM's macro-level validation, we introduce the Hierarchical Feature Calibration (HFC) module to ensure micro-level fidelity across clinically relevant retinal depths. Recognizing the distinct textural characteristics inherent to different retinal layers, HFC performs layer-specific refinement to preserve both structural accuracy and textural authenticity. This targeted calibration strategy enhances the resolution of anatomical landmarks and microvascular details, significantly improving diagnostic quality for complex retinal pathologies.

We trained two specialized models for layer-specific projection synthesis: $G_{ILM-OPL}$ generates projections from the Internal Limiting Membrane (ILM) to Outer Plexiform Layer (OPL), while $G_{OPL-BM}$ produces projections from OPL to Bruch's Membrane (BM). Each HFC module utilizes a UNet backbone (Ronneberger et al., 2015) for projection-to-projection transformation.

To enhance spatial coherence and suppress artifacts, we incorporate Total Variation Loss, effectively reducing noise while preserving edge information. This combined approach ensures precise content matching while maintaining smooth, natural feature transitions, yielding superior visual quality and authenticity.

Let $x_{OCTA}^{proj}$ and $\widehat{x}_{OCTA}^{proj}$ denote z-axis averaged projection images from real and generated 3D OCTA volumes, respectively. The hierarchical loss formulation comprises:

$$\mathcal{L}_{proj} = \left\| x_{OCTA}^{proj} - \widehat{x}_{OCTA}^{proj} \right\|_1 \tag{7}$$

$$\mathcal{L}_{ILM-OPL} = \left\| G_{ILM-OPL}(x_{OCTA}^{proj}) - G_{ILM-OPL}(\widehat{x}_{OCTA}^{proj}) \right\|_1 \tag{8}$$

$$\mathcal{L}_{OPL-BM} = \left\| G_{OPL-BM}(x_{OCTA}^{proj}) - G_{OPL-BM}(\widehat{x}_{OCTA}^{proj}) \right\|_1 \tag{9}$$

$$\mathcal{L}_{tv3d} = \sum_{d,h,w}^{D-1,H,W} |V_{d+1,h,w} - V_{d,h,w}| + \sum_{d,h,w}^{D,H-1,W} |V_{d,h+1,w} - V_{d,h,w}|$$
$$+ \sum_{d,h,w}^{D,H,W-1} |V_{d,h,w+1} - V_{d,h,w}| \tag{10}$$

$$\mathcal{L}_{HFC} = (\mathcal{L}_{proj} + \mathcal{L}_{ILM-OPL} + \mathcal{L}_{OPL-BM} + \mathcal{L}_{tv3d}) \times \lambda_B \tag{11}$$

We pre-train VSM and HFC modules independently, then freeze their parameters during complete framework training. The total loss for OCT-to-OCTA synthesis is formulated as:

$$\mathcal{L}_{octa\_loss} = \mathcal{L}_{3dGAN} + \lambda_C \times \mathcal{L}_{VSM} + \mathcal{L}_{HFC} \tag{12}$$

For the reverse OCTA-to-OCT transformation, we employ the identical framework architecture while substituting the generator backbone with ResUNet (Diakogiannis et al., 2020) to better capture structural details. The corresponding loss function is:

$$\mathcal{L}_{oct\_loss} = \mathcal{L}_{Adv3d} + \lambda_{A'}\mathcal{L}_{L13d} \tag{13}$$

## 4 EXPERIMENTS

### 4.1 DATASETS

For our experiments, we utilize the publicly available OCTA-500 (Li et al., 2024a) dataset. This dataset is divided into two subsets, OCTA-3M and OCTA-6M, according to the field of view (FoV) and resolution of the scans. Our work specifically utilizes the larger OCTA-6M subset.

This subset is ideal for our task as it provides 300 co-registered pairs of structural OCT and functional OCT Angiography (OCTA) volumes. The OCT volumes provide detailed cross-sectional information of the retinal layers, while the corresponding OCTA volumes visualize the retinal vascular network. For all experiments, we adhered to the official data partition for the training, validation, and test sets. A comprehensive summary of the dataset specifications, including detailed volume dimensions and data splits for both subsets, is provided in the Appendix A.1(Table 7).

### 4.2 EXPERIMENTAL DETAILS

We employed standard image quality metrics: Mean Absolute Error (MAE), Peak Signal-to-Noise Ratio (PSNR), and Structural Similarity Index (SSIM) (Wang et al., 2004).

Our framework comprises three integrated components for reliable cross-modal retinal image conversion. The core WaveAtten3D-GAN extends pix2pix with wavelet transforms, multi-scale spatial attention, and adaptive discriminator augmentation for accurate 3D reconstruction. The Vessel Structure Matcher (VSM) module, based on IPN-V2 (Li et al., 2024a), preserves vascular morphology. The Hierarchical Feature Calibration (HFC) module employs UNet (Ronneberger et al., 2015) for depth-level projection conversion, ensuring structural fidelity for clinical evaluation.

Training utilized Adam optimizer for 200 epochs on an NVIDIA RTX 4090 GPU (24GB). MAE served as the primary validation metric. Detailed training parameters for each component are provided in Appendix A.2(Table 8).

We conducted comprehensive visual evaluations of our bidirectional synthesis framework. Detailed visualization results including B-scan comparisons and projection map analyses for both OCT-to-OCTA and OCTA-to-OCT conversions are presented in Appendix A.3 (Figure 4 and Figure 5).

### 4.3 COMPARISON EXPERIMENTS

We compared our method with baseline models (p2pGAN 2D/3D) (Isola et al., 2017), OCT conversion methods (Adjacent GAN (Li et al., 2020), 9B18CN UNet (Lee et al., 2019), TransPro (Li et al., 2024b)), and Palette (Saharia et al., 2022). Table 1 presents the results, where ↑ indicates higher-is-better and ↓ indicates lower-is-better metrics.

Table 1: Comparative evaluation of synthesis quality metrics against baseline and state-of-the-art approaches.

| Model | PSNR↑ | SSIM↑ | MAE↓ |
|-------|-------|-------|------|
| p2pGAN 2D | 27.65 | 87.15 | 0.0995 |
| p2pGAN 3D | 29.77 | 88.18 | 0.0217 |
| Adjacent GAN | 28.05 | 85.03 | 0.1021 |
| 9B18CN UNet | 27.91 | 83.69 | 0.1135 |
| Palette | 30.02 | 87.13 | 0.0881 |
| TransPro | 30.53 | 88.35 | 0.0854 |
| Ours | **30.58** | **90.64** | **0.0199** |

## 4.4 ABLATION STUDIES

We evaluated VSM and HFC contributions through ablation studies (Table 2). VSM excels by explicitly preserving 3D vascular morphology, while HFC ensures hierarchical feature consistency. Their combined application yields optimal performance with additive effects.

Table 2: Component-wise ablation study evaluating VSM and HFC contributions.

| Model | VSM | HFC | PSNR↑ | SSIM↑ | MAE↓ |
|---|---|---|---|---|---|
| WaveAtten3D-GAN | – | – | 30.4019 | 90.30 | 0.0206 |
| w/ VSM | ✓ | – | 30.4988 | 90.49 | **0.0199** |
| w/ HFC | – | ✓ | 30.4476 | 90.57 | 0.0201 |
| w/ Both | ✓ | ✓ | **30.5832** | **90.64** | **0.0199** |

L1 loss weighting analysis (Table 3) reveals that $\lambda_A = 120$ optimally balances structural integrity with detail preservation, while excessive weighting causes over-smoothing.

Table 3: Quantitative evaluation of $\lambda_A$ parameter selection.

| $\lambda_A$ | PSNR↑ | SSIM↑ | MAE↓ |
|---|---|---|---|
| 100 | 30.2338 | 89.96 | 0.0206 |
| **120** | **30.4019** | **90.30** | **0.0206** |
| 130 | 30.3067 | 90.24 | 0.0205 |

VSM loss weight optimization (Table 4) demonstrates that $\lambda_C = 5$ achieves optimal vascular morphology preservation without compromising overall fidelity.

Table 4: Ablation study on $\lambda_C$ hyperparameter at optimal $\lambda_A$ setting.

| $\lambda_A = 120, \lambda_C$ | PSNR↑ | SSIM↑ | MAE↓ |
|---|---|---|---|
| 3 | 30.2748 | 90.24 | 0.0206 |
| **5** | **30.4988** | **90.49** | **0.0199** |
| 7 | 30.1599 | 90.08 | 0.0209 |

For OCTA-to-OCT conversion, optimal L1 weight $\lambda_{A'} = 15$ provides best performance (Table 5).

## 4.5 CLINICAL DISEASE CLASSIFICATION

To evaluate the clinical utility of our synthesized images, we conducted disease classification experiments using real and synthesized modalities. We trained a ResNet-34 classifier on four different input configurations: (1) real OCT images only, (2) real OCTA images only, (3) real OCT combined with synthesized OCTA, and (4) synthesized OCT combined with real OCTA. The classifier was trained to distinguish between healthy and diseased retinal conditions.

Table 6 presents the classification performance metrics. The results demonstrate that combining real OCT with our synthesized OCTA images (configuration 3) achieves the highest accuracy of 75.06%, surpassing both single-modality baselines. This improvement suggests that our synthesized OCTA effectively complements structural OCT information with functional vascular details. Similarly, the combination of synthesized OCT with real OCTA (configuration 4) maintains competitive performance at 74.66% accuracy, validating the bidirectional synthesis capability of our framework.

Notably, the multi-modal approaches consistently outperform single-modality baselines, with real OCT + synthesized OCTA achieving a 7.1% improvement over OCT alone and 29.2% over OCTA alone in accuracy. The high specificity (95.7%) achieved by synthesized OCT + real OCTA demonstrates excellent capability in identifying healthy cases, while the high sensitivity (70.2%) of real OCT + synthesized OCTA indicates strong disease detection performance. These results validate that our synthesis framework generates clinically meaningful representations that enhance diagnostic capability when combined with real imaging modalities.

Table 5: Quantitative evaluation of $\lambda_{A'}$ parameter selection.

| $\lambda_{A'}$ | PSNR↑ | SSIM↑ | MAE↓ |
|---|---|---|---|
| 5 | 26.5979 | 37.80 | 0.0679 |
| 10 | 26.4424 | 37.13 | 0.0688 |
| **15** | **27.1163** | **39.42** | **0.0652** |

Table 6: Disease classification performance using real and synthesized modalities (95% CI in brackets).

| Input Modality | Accuracy | Precision | Recall | F1-Score | Sensitivity | Specificity |
|---|---|---|---|---|---|---|
| Real OCT | 0.701 [0.583-0.817] | 0.875 [0.743-0.971] | 0.667 [0.523-0.811] | 0.755 [0.638-0.857] | 0.667 [0.523-0.811] | 0.778 [0.571-0.947] |
| Real OCTA | 0.582 [0.450-0.700] | 0.903 [0.765-1.000] | 0.451 [0.308-0.605] | 0.598 [0.456-0.735] | 0.451 [0.308-0.605] | 0.887 [0.727-1.000] |
| Real OCT + Synth. OCTA | **0.751** [0.700-0.797] | 0.919 [0.873-0.960] | **0.701** [0.635-0.761] | **0.798** [0.750-0.843] | **0.702** [0.642-0.770] | 0.857 [0.776-0.922] |
| Synth. OCT + Real OCTA | 0.747 [0.700-0.793] | **0.971** [0.940-0.993] | 0.655 [0.596-0.718] | 0.782 [0.738-0.825] | 0.655 [0.591-0.719] | **0.957** [0.910-0.990] |

## 5 CONCLUSION

We present a comprehensive wavelet-enhanced deep learning framework for bidirectional OCT-OCTA image conversion that enables true multi-modal synergistic analysis from single-modality acquisitions. By integrating a Vessel Structure Matcher with a Hierarchical Feature Calibration mechanism, our approach ensures both morphological accuracy and textural fidelity essential for clinical diagnostics.

This bidirectional framework addresses critical clinical needs by enabling comprehensive diagnostic workflows that leverage the complementary strengths of both modalities. The OCT-to-OCTA pathway democratizes access to vascular imaging in resource-constrained settings, while the OCTA-to-OCT pathway provides essential structural context for vascular findings, facilitates artifact disambiguation, and enables cross-modal validation when original imaging is compromised. This integrated multi-modal analysis capability represents a significant advancement in precision ophthalmology, allowing clinicians to extract maximum diagnostic information from available imaging resources.

Our experimental results demonstrate state-of-the-art synthesis performance, and clinical validation confirms that synthesized modalities enhance diagnostic accuracy by up to 29% compared to single-modality baselines. This improvement underscores the clinical significance of multi-modal synergistic analysis, where the integration of structural and functional information leads to more comprehensive and accurate disease assessment.

Future work will focus on architectural refinement, exploration of diffusion-based generative models, and multi-center clinical validation to establish real-world efficacy and accelerate clinical adoption.

### ACKNOWLEDGMENTS

We thank the creators of the OCTA-500 dataset for making their data publicly available.

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

## A APPENDIX

### A.1 DATASET DETAILS

Table 7: Overview of the OCTA-500 Dataset

| Access Link | `https://ieee-dataport.org/open-access/octa-500` | |
|---|---|---|
| **Attribute** | **OCTA-3M Subset** | **OCTA-6M Subset** |
| Total Volumes | 200 pairs | 300 pairs |
| Field of View (FoV) | 3mm $\times$ 3mm $\times$ 2mm | 6mm $\times$ 6mm $\times$ 2mm |
| Volume Size | 304px $\times$ 304px $\times$ 640px | 400px $\times$ 400px $\times$ 640px |
| Projection Map Size | 304px $\times$ 304px | 400px $\times$ 400px |
| **Dataset Split** | | |
| Training Set | 140 volumes | 180 volumes |
| Validation Set | 10 volumes | 20 volumes |
| Test Set | 50 volumes | 100 volumes |

### A.2 TRAINING PARAMETERS

Table 8: Detailed Training Parameters for Each Component

| Component | Parameter | Value |
|---|---|---|
| WaveAtten3D-GAN | Optimizer | Adam |
| | Learning Rate | $2 \times 10^{-4}$ |
| | Batch Size | 1 |
| | Epochs | 200 |
| | Loss Weights | $\lambda_A = 120$ (OCT$\rightarrow$OCTA) |
| | | $\lambda_{A'} = 15$ (OCTA$\rightarrow$OCT) |
| VSM (IPN-V2) | Optimizer | Adam |
| | Learning Rate | $1 \times 10^{-4}$ |
| | Epochs | 150 |
| | Loss Function | Cross-entropy |
| | Loss Weight | $\lambda_C = 5$ |
| HFC (UNet) | Optimizer | Adam |
| | Learning Rate | $1 \times 10^{-2}$ |
| | Epochs | 100 |
| | Loss Function | L1 + TV |
| | Loss Weight | $\lambda_B = 0.25$ |
| ResNet-34 Classifier | Optimizer | Adam |
| | Learning Rate | $1 \times 10^{-3}$ |
| | Batch Size | 16 |
| | Epochs | 50 |

### A.3 VISUAL RESULTS

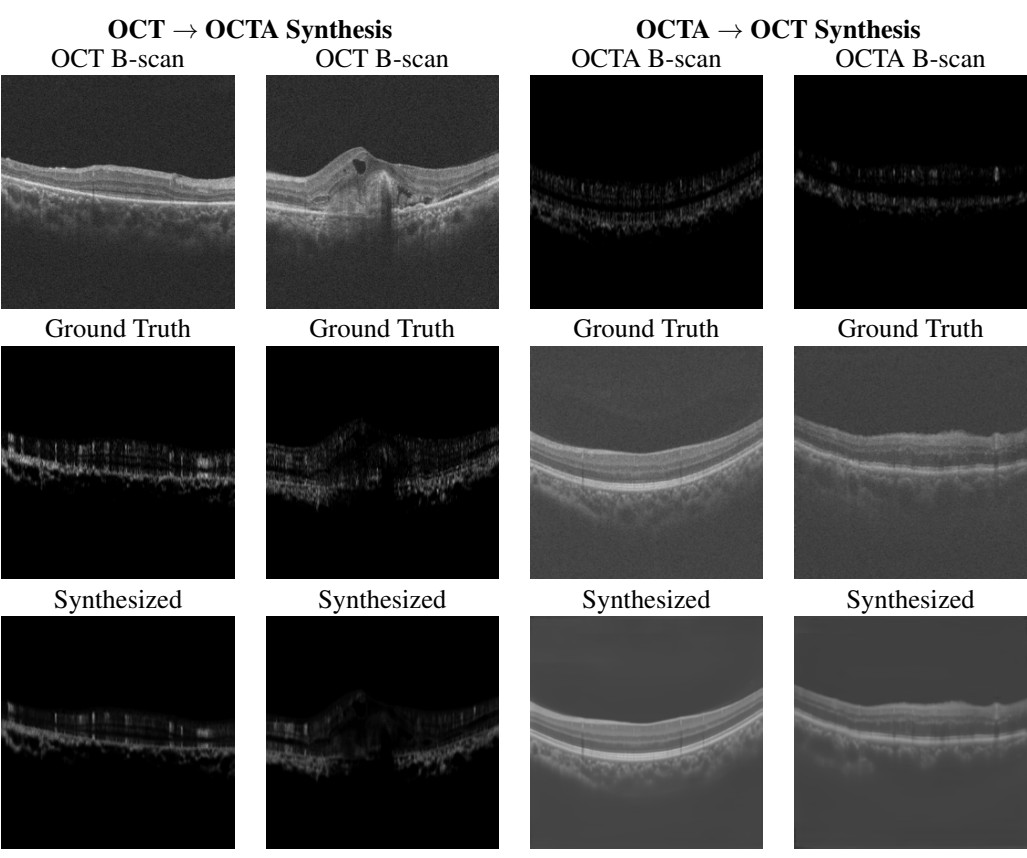

Figure 4: Bidirectional B-scan synthesis results. Left: OCT-to-OCTA conversion showing structural OCT input (row 1), OCTA ground truth (row 2), and synthesized OCTA (row 3). Right: OCTA-to-OCT conversion showing OCTA input (row 1), OCT ground truth (row 2), and synthesized OCT (row 3).

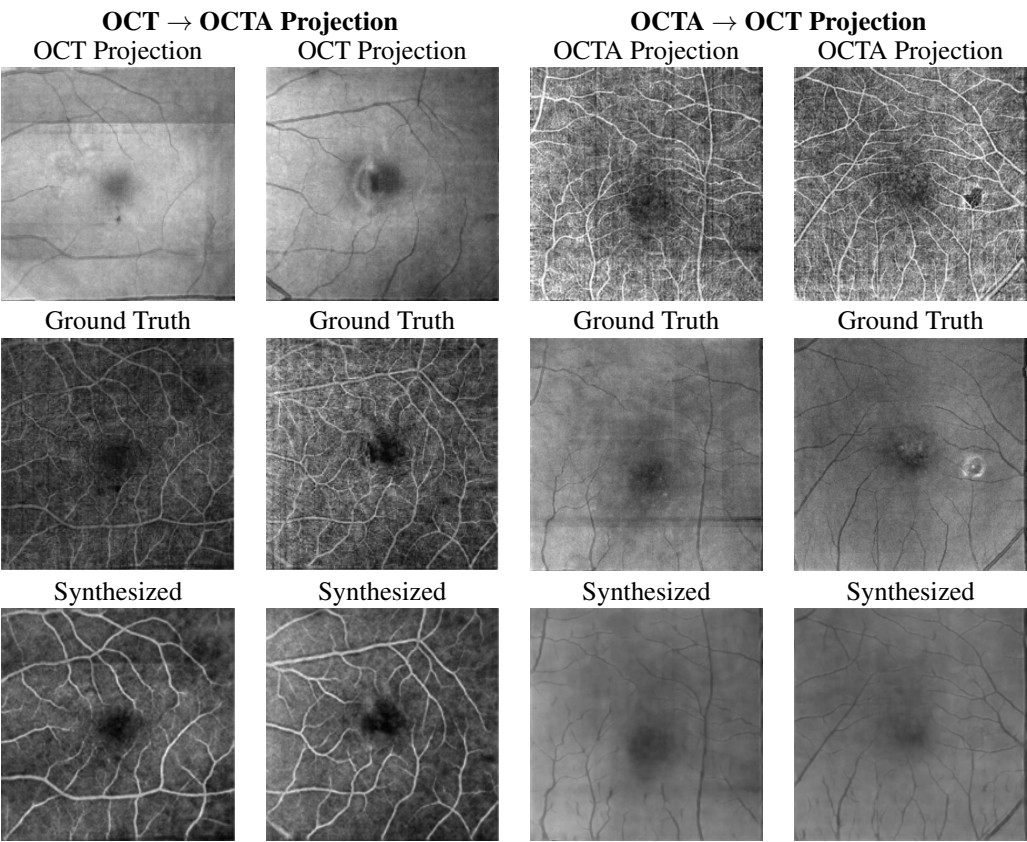

Figure 5: Bidirectional projection map synthesis results. Left: OCT-to-OCTA conversion showing OCT projection input (row 1), OCTA projection ground truth (row 2), and synthesized OCTA projection (row 3). Right: OCTA-to-OCT conversion showing OCTA projection input (row 1), OCT projection ground truth (row 2), and synthesized OCT projection (row 3).

