# OpenReview forum: "Vascular Information-Guided Automated Conversion Between OCT and OCTA"
_ICLR.cc/2026/Conference — ICLR 2026 Conference Withdrawn Submission_

### Official Review · Reviewer_TRGb · 2025-10-25

**Soundness:** 3
**Presentation:** 3
**Contribution:** 2
**Rating:** 2
**Confidence:** 4

**Summary:**

The paper presents a wavelet enhanced deep learning framework for bidirectional conversion between OCT and OCTA

**Strengths:**

The designed framework is complete.

**Weaknesses:**

con:

The paper does not have novlety in machine leanring, maybe consider it as an application.

Experiement is too simple, please consider to test more datasets.

**Questions:**

Maybe more data to test so we can have a complete discussion about the method.

---

### Official Review · Reviewer_RPw9 · 2025-10-28

**Soundness:** 3
**Presentation:** 3
**Contribution:** 3
**Rating:** 6
**Confidence:** 5

**Summary:**

This paper starts from bidirectional OCT-to-OCTA generation for considering the inverse transformation’s clinical value. In this regard, the authors propose the first comprehensive architecture for both OCT-to-OCTA and OCTA-to-OCT transformation, enabling true multi-modal synergistic analysis from single-modality inputs.

**Strengths:**

1. This paper provides an interesting perspective, i.e.,  bidirectional OCT-to-OCTA generation for considering the inverse transformation’s clinical value.
2. Although the proposed method seems to be incremental from existing works, the integration of the projection-to-projection transformation is novel to me.
3. The disease classification experiments are informative and helpful to verify the quality of the generated images.
4. Codes are available.

**Weaknesses:**

1. The vessel segmentation models may be hard to access for a new scenario.
2. The details of how to achieve bidirectional OCT-to-OCTA generation are unclear. More details are necessary.

**Questions:**

1. What is the composition of the test set in the CLINICAL DISEASE CLASSIFICATION tasks? Does it only contain OCT or OCTA real images?
2. What is the performance of using only synthesized OCT or OCTA?

---

### Official Review · Reviewer_KPh4 · 2025-10-31

**Soundness:** 2
**Presentation:** 3
**Contribution:** 3
**Rating:** 4
**Confidence:** 4

**Summary:**

This paper presents a bidirectional OCT <-> OCTA image conversion framework, utilizing a 3D wavelet-based transformer with the Vessel Structure Matcher (VSM) and Hierarchical Feature Calibration (HFC) techniques. The framework achieves high quantitative metrics.

**Strengths:**

1) The clinical disease classification shows that the synthetic images can improve the performance of the downstream classification task.
(2) The paper evaluates the proposed framework on a public dataset

**Weaknesses:**

(1) When analysing the quality of the synthetic images, the authors only use PSNR, SSIM, and MAE. Perceptual and structural metrics are needed for comprehensive evaluation.
(2) The paper lacks qualitative analysis. Some examples of the synthetic images are needed to demonstrate the performance of the framework.

**Questions:**

The authors may consider adding safety analysis to the proposed framework. Using the proposed framework, will there be any risk of false positives/negatives in the downstream classification task?

---

### Official Review · Reviewer_7qwq · 2025-11-03

**Soundness:** 2
**Presentation:** 2
**Contribution:** 1
**Rating:** 2
**Confidence:** 5

**Summary:**

This paper describes a generative framework for converting optical coherence tomography (OCT) images to and from OCT-angiography (OCTA). This is accomplished by three modules, a GAN-like model, a feature matching model, and a calibration model. The backbone models are all taken from well-known established architectures such as U-NET encoder/decoder models, 3D wavelet transforms, and other segmentation tools. In addition to the lack of originality and novelty, the paper has a multitude of shortcomings that makes it unsuitable for acceptance.

1) Lack of novelty: The paper claims that the architectural innovations is in integration of wavelets with attention guided reconstruction. However, there is no clear outline of how this is achieved, or a clear theoretical support for its ability to provide optimal solutions. In addition, the tools used to accomplish this integration are rather well established in their own domains,  e.g. Wavelets from (Phung, et al. 2023), multi-scale attention from (Ouyang et al., 2023). The generative model is also a well known encoder/decoder. The pretraining model used for vessel structure matching is also from (Li et al. 2024a).

2) Theoretical issues: The paper has major issues with theoretical soundness of its claims. The paper purports to have developed a novel bi-directional synthesis framework. However, the technical details of the paper do not present a clear and theoretical framework for its bi-directionality. Rather, it appears that the bi-directionality is achieved by inputting OCT images to produce OCTA, and then inputting OCTA to produce OCT. This is still a uni-directional approach, and far from achieving a truely bi-directional model. Moreover, the theory behind the utility of wavelet-guided features are not developed. For example, how does one distinguish between high-frequency features that are due to micro structures, as opposed to pathological anomalies. There is also no theoretical details on how the morphological preservation is achieved. The paper also claims that its approach is multi-model. However, both OCT and OCTA data are 3D structures of the same modality. This is a grave misrepresentation of the core contribution of the paper.

3) Unsupported and false claims: There are many unsupported claims (either by theory or empirically). For example, integration of wavelets enables morpholoigical coherence. This claim is neither supported by theory or empirically. The claim that this approach is bidirectional is completely false. The claim that the approach is multi modal is also completely false. The claim that the multi-scale attention module selectively emphasizes clinically relevant structures is also completely unsupported. Claim that wavelets maintain microvascular structures and other global and pathological features is also not supported. It is important to note that many structural and pathological features have representations in multiple frequency and spatial bands. There is no theoretical foundation about that supports the decompositional disentaglement purported here.

**Strengths:**

None noted.

**Weaknesses:**

Presented in the summary.

**Questions:**

None

---

### Note · Authors · 2025-11-24

I have read and agree with the venue's withdrawal policy on behalf of myself and my co-authors.